# Towards Facial Biometrics for ID Document Validation in Mobile Devices

**Iurii Medvedev** [1,*] **, Farhad Shadmand** [1] **, Leandro Cruz** [1,2] **and Nuno Gonçalves** [1,3]

1   Institute of Systems and Robotics, University of Coimbra, R. Silvio Lima, 3030-194 Coimbra, Portugal; farhad.shadmand@isr.uc.pt (F.S.); l.cruz@psenterprise.com (L.C.); nunogon@deec.uc.pt (N.G.)
2   Siemens Process Systems Engineering, London W6 7HA , UK
3   Portuguese Mint and Official Printing Office (INCM), 1000-042 Lisbon, Portugal
*   Correspondence: iurii.medvedev@isr.uc.pt

**Abstract:** Various modern security systems follow a tendency to simplify the usage of the existing biometric recognition solutions and embed them into ubiquitous portable devices. In this work, we continue the investigation and development of our method for securing identification documents. The original facial biometric template, which is extracted from the trusted frontal face image, is stored on the identification document in a secured personalized machine-readable code. Such document is protected from face photo manipulation and may be validated with an offline mobile application. We apply automatic methods of compressing the developed face descriptors to make the biometric validation system more suitable for mobile applications. As an additional contribution, we introduce several print-capture datasets that may be used for training and evaluating similar systems for mobile identification and travel documents validation.

**Keywords:** artificial neural networks; biometrics; document handling; face recognition





## 1. Introduction

Document security has been an important issue since the appearance of the first documents and banknotes. Physical documents are still the first and ultimate authentication method and that is why their protection against spoofing attacks is important. Since the face image is one of the most important biometric components of ID and travel documents, its security is a prominent concern for official issuing organizations [1].

Face recognition techniques have been drawing a lot of attention in the last decade and, particularly, with the development of deep learning tools such as convolutional neural networks, they achieve outstanding accuracy levels.

Facial recognition technology impacts the overall security, allowing to automate ID document validation. The pipeline of this process usually follows the differential scenario, which implies that, during the verification procedure, the trusted reference is available. As a source of this reference, various systems may use the face image, a template that is previously stored and secured, or the trusted live captured image of a person from the border control camera.

The scenario of 1-1 authentication (verification) is a form of identity validation of a tested individual. At this scale, it is not required to store the remote biometric template/samples database, which eliminates the risks related to identity database theft or fraud at the point of control. The limitation of such a scenario provides face recognition systems with a number of benefits. They can be convenient and safe in applications, such as accessing the security area of personal devices or proving the identity during automated border control when trusted live capture is compared with the face image printed in the passport.

The particular case of 1-1 authentication is the so-called match-on-document scenario, which assumes that the trusted and secured biometric template is stored on the documents

themselves. This strategy allows performing document validation in an offline mode to reduce the information security risks when storing or accessing databases of face images and templates are not allowed. The approach we discuss in this paper is directed to the applications for this scenario.

Various face recognition systems usually follow a 1-N authentication scenario which has some peculiarities but which is indeed less relevant to this work.

Face recognition systems are threatened by presentation attacks (spoofing attacks) of different kinds. In general, they usually intend to disguise the real subject identity or deceive the system to be impersonated as a different identity [2]. Such impostor attacks can lead to the gaining of illegitimate access by the fraudulent user and pose significant threads for security fields in companies and government sectors [3].

In applications of facial biometric recognition technologies in security systems for ID documents, several aspects are relevant. These solutions tend to become embedded in portable devices, such as smartphones. The specific tendency of certain solutions is to accomplish the offline validation of documents, to avoid risk related to compromising the network connectivity at any level. In this case, the minutia information is extracted from sources of biometric data (faces, fingerprints, iris, etc.), without storing or accessing databases of face images/templates.

The face recognition task for document security is usually constrained by considering frontal face images (in accordance with the ISO/IEC 19794-5 [4]). Since our method is focused on protecting ID and travel documents, we follow that statement. However, it is worth noting that, conceptually, instead of the frontal face image, any other source of biometric data (e.g., fingerprint pattern or iris) may be used in the design of such a validation system.

The designed biometric validation system secures the ID document by extracting the template from an enrolled frontal face image and encoding it into an MRC, which is further printed on the document in the specified area (Figure 1). The choice of MRC and encoding strategy is provided by the requirement that the bona fide MRC cannot be generated by an unauthorized issuer, which complicates producing fraudulent identification documents for deceitful impersonating.

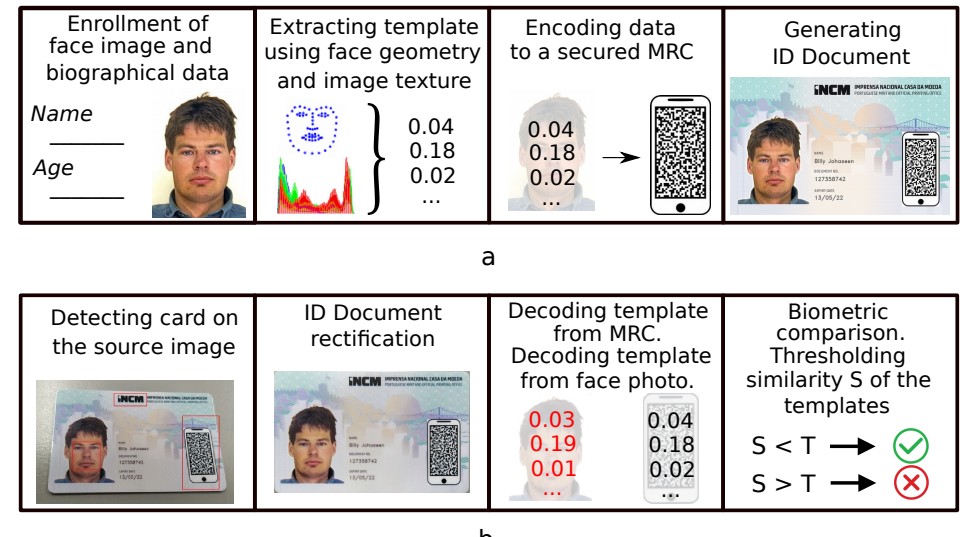

**Figure 1.** Pipeline of the proposed system: (**a**) ID card generating; and (**b**) ID card validating.

The validation of such document is performed by extracting two biometric templates from the frontal face photo and the MRC which are printed on its exterior. The templates are compared to determine if they belong to the same identity. The pipeline of the process in such a match-on-document scenario does not require network access, and the validation may be performed totally offline. However, the proposed approach in practical applications

indeed may be expanded straightforwardly by including biographic data (name, date of birth, etc.) into the MRC.

It should be pointed out that the formulation of the face recognition problem in our method is specific. The main task being solved is to protect the particular instance of the issued document at the moment of its personalization with variable data. With this formulation, we do not focus on matching several photos of a single identity, although the designed method may be expanded for deployment in such a scenario. That is stated due to the fact that any newly issued document will contain the secured encoded biometric template, which corresponds to the particular face image to be printed on this document. However, in the scope of this work, the mentioned limitation is provided only by the choice of the training and test datasets. That is why we clarify the formulation of a problem as the protection of the particular face image of the ID document from various biometric impostor attacks.

Furthermore, one must notice that the matching between the two templates is not expected to be a perfect one, since the minutiae extracted from the frontal face are always different each time the validation occurs, due to image color and radiometric distortion, lighting conditions and the camera's intrinsic and extrinsic parameters, thus complicating the problem of matching different identities.

Summing up, the research and development of security solutions for match-on document scenario is important and for instance have been performed in our initial study [5]. In this work, which is the extended version of the paper presented at the BIOSIG2020 conference, we continue this investigation towards a compact offline mobile solution to secure ID and travel documents. By employing the secured MRC to embed the ID document with a carefully designed facial descriptor, we perform the document validation without storing biometric samples and templates in the remote database.

In comparison with the previous work, we modify the facial biometric template by including texture components. Our implementation of the facial descriptor is compact and optimized for usage in mobile devices. We also estimate the effect of template compression following the concept of match-on-document verification. Finally, we present several collected print/capture datasets which may be useful for analyzing the performance.

The paper is organized in the following way. In Section 2, we review some recently published works related to our research. Section 3 represents the discussion of the improved face descriptor implementation approach of differential validating and compressing the designed descriptors. Section 4 is devoted to the choice and application of machine-readable code (MRC) to our work. In Section 5, we present the details of the acquired in-house datasets. The discussion on the experimental results is performed in Section 6.

## 2. Related Work

The conventional pipeline of processing the input digital image (which may be acquired in different ways) in face recognition systems usually includes face detection [6] with optional alignment [7], followed by the face representation [8]. The last stage may be generally formulated as transforming the preprocessed face image into the low-dimensional feature space where various recognition tasks can be easily performed. This recognition scenario is usually defined by the practical purposes of the system. Intense efforts have been expended for the search of a better feature domain that possesses high face discriminative power and enough separability for distinguishing images corresponding to disjoint identities.

### 2.1. AAM in Face Recognition

Significant improvements in engineered methods for face recognition were achieved with the development of various techniques for automatic detection of special landmarks that allow localizing semantic regions on the face image. The list of selected face features is usually included in the standardized active appearance model (AAM) [9]. Such an

approach gives huge opportunities for analyzing the face structure and processing the raw face images which have become very useful for face recognition applications.

For example, Abdulameer et al. [10] used facial features extracted with the use of AAM for the purposes of face verification that was performed with the trained classifier. Ouarda et al. [11] analyzed geometric face distances and characteristics of the semantic face shapes for face recognition purposes. The face recognition method reported by Juhong et al. [12] is based on face geometric invariances.

Another approach to face recognition with the use of an active appearance model is based on detecting semantic regions and extracting local texture features for further analysis. For instance, Ahonen et al. [13] considered both shape and texture information to discriminate face images. The face descriptor in this method is based on Local Binary Pattern (LBP) histograms extracted from the partitioned image. The dissimilarity metric between the descriptors is estimated by the nearest neighbor classifier.

Many improvements were introduced to this technique. For instance, Shen et al. [14] adopted discriminative LBP features for different color channels to enhance the performance for images with severe changes in illumination and face resolutions.

Another popular technique for face recognition that deals with image texture is the histogram of oriented gradients (HOG) method, which usually implies sub-sampling images to small blocks and counting proportions of gradient orientation in these localized segments of an image. The extracted coefficients may further serve as discriminative features of the image.

The example of such an approach was reported by Shu et al. [15] who analyzed different factors that affect the HOG-based face descriptor and the performance and computational efficiency in comparison with other texture-based techniques.

Deniz et al. [16] considered the HOG descriptor with sub-sampling based on the facial landmarks. Various methods for increasing the robustness of extracted HOG features were considered by analyzing the impact of facial feature location error and replacing the detected features with the regular grid.

Other texture analysis techniques are generally less popular but also attract the interest of face recognition research. The Gabor feature method is widely used in computer vision for pattern recognition tasks. Applying special Gabor filters, it is possible to extract the directivity and frequency of the content within the vicinity of some point or region. For example, Yunfeng et al. [17] used concatenated Gabor wavelet coefficients, which are extracted in the vicinity area of each detected facial landmark. Further, the descriptors are distinguished with a support vector machine classifier.

In [5], the face recognition solution for the match-on-document scenario is introduced. It employs the process of encoding the biometric template into the secured MRC to be printed on the document. By comparison with this work, we revisit and improve the implementation of the facial biometric template, considering the methods of its compressing, which are important for the target platform (mobile devices).

### 2.2. Face Recognition with Machine Learning

Modern face recognition intensively uses recent achievements in machine learning. These tools are usually served to learn the discrimination of face descriptors, for example by solving a classification task to estimate the similarity between images. The face representation in these approaches is usually engineered and based on low-level face image information [10,13].

Another approach to face recognition came with convolutional deep networks which give the ability to efficiently learn discriminative face features themselves even from unconstrained images. These learned features generally do not include local image characteristics and realize the face description in a high-level manner [8]. Significant popularity in face recognition systems was gained by metric learning approaches which are focused on straightforwardly optimizing the face representation. For example, Schroff et al. [18] introduced a triplet loss for face recognition which minimizes the distance in the feature

domain between samples of the same identity and maximizes it for the disjoint identities. However, such methods usually require unreasonably large datasets accompanied by a carefully designed sample mining strategy for successful convergence.

The best performance in unconstrained face recognition is achieved by methods that consider the problem as a classification task separating images by their identities. These approaches usually utilize softmax-based classification loss [8] which allows learning the discriminative face features, which may be further used for distinguishing tasks with trivial similarity metrics. Nowadays, investigation is focused on modifying softmax loss by different means. The main purpose of most of the published improvements is enlarging the inter-class variance and reducing the inter-class compactness [19]. For example, particular attention was paid to constraining the target feature distribution with a margin of a different kind [20,21].

Some deep learning methods also find their application in face recognition systems for document security. As an example, Shi et al. [22] proposed a method for 1-1 authentication for the differential automatic border control scenario. In their system, the face photo on the ID document is validated with the help of life face capture. The two images are processed by separate networks to estimate the similarity of their deep representations.

However, the intricacy and the lack of a clear sense of extracted deep representations may be an obstacle in practice while embedding these approaches in portable devices. The computational complexity requirements of deep learning approaches are still high for most of the smartphones on the market.

At the same time, the differential manner of match-on-document scenario implies 1-1 verification of templates extracted from an original digital image and its copy, which is printed on the ID document. With such a scenario, the engineered features, which may catch characteristic peculiarities of the particular image, have better usage perspectives than learned high-level features. However, machine learning tools indeed may be applied for distinguishing such engineered face feature representations.

### 2.3. Industry Solutions

Industry solutions also follow the advances in facial recognition for protecting ID and travel documents with modern techniques. Two noticeable products were developed by Jura (Digital IPI) [23] and IDEMIA (Lasink and DocSeal) [24]. These approaches attempt to embed face recognition systems into ubiquitous portable devices (e.g., smartphones). Such validation solutions may broadcast the convenience of authenticity verification of documents and products, while at the same time allowing to reduce the requirements for the sophisticated equipment. Their main idea is to augment the document with printed elements that store encoded personalized data to be further extracted and decoded with the use of a portable device with a digital camera.

In our work, we adopt a similar approach for protecting ID and travel documents. However, the above methods are included in proprietary commercial systems and are not publicly available, which does not allow completing benchmarks and comparison. That is why existing benchmarks (e.g., the NIST FRVT challenge [25]) have some submission restrictions and usually accept the solutions in the compiled form without the requirements of submitting the source code.

## 3. Materials and Methods

In the scope of this work, we followed the motivation of developing a simple and compact method for offline document validation that includes an in-house solution for face description and their verification. We also followed the trends of modern validation applications that rely on biometric recognition and emphasized portability and ubiquity.

### 3.1. Facial Biometric Template

In our investigation, we focus on the search for a method for facial description which is optimized for encoding into MRC and embedding into mobile devices. We combine

our facial biometric template from several types of discriminative features which include information about face geometry and texture.

The process of extracting features from the frontal face image starts with the applying active appearance model and detecting facial landmarks (Figure 2). We employ the standard appearance model that includes 68 facial landmarks and entirely specifies face semantic regions to be further processing by the algorithm.

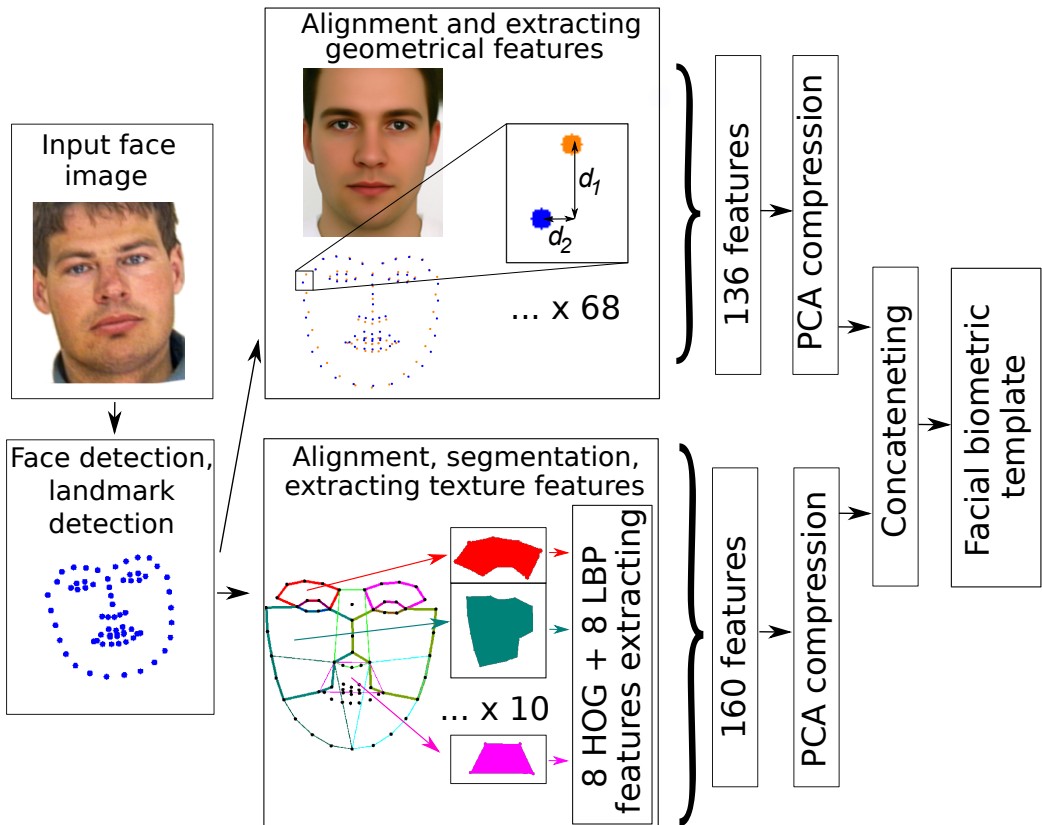

**Figure 2.** The process of extracting facial biometric template from the input face image.

The coordinates of the detected set of landmarks carry all discriminative geometric information given by the chosen model. However, due to the uncertainty of face parameters (e.g., size or pose) on the source image, these raw data cannot be directly used for making biometric comparisons and require some normalization procedure.

To achieve that, we define some fixed set of landmarks that serves as a reference for alignment. In that case, if two sets of landmarks from different arbitrary face images are aligned to the defined supporting set, they also become aligned to each other. In our work, we choose this supporting set by extracting facial landmarks from averaged face image in Figure 2.

The alignment of the input set of points $\{x_i, y_i\}$ with the supporting set, is implemented as a coordinate transformation to $\{x'_i, y'_i\}$ by rotating, scaling and shifting operations, defined by Equation (1). The $\alpha$ rotation is determined to achieve the horizontal face pose. Scaling is performed by the relation of values of face contour perimeters ($P_{sup}$ corresponds to the supporting set and $P$ to the input set), which is the selection of points with indices in the range [0–26] (depicted with blue color in Figure 2). The shift $S(s_x, s_y)$ is defined as the difference between the centers of supporting and scaled input sets of points.

$$\begin{bmatrix} x'_i \\ y'_i \end{bmatrix} = \frac{P_{sup}}{P} * \begin{bmatrix} cos(\alpha) & -sin(\alpha) \\ sin(\alpha) & cos(\alpha) \end{bmatrix} * \begin{bmatrix} x_i \\ y_i \end{bmatrix} + \begin{bmatrix} s_x \\ s_y \end{bmatrix} \tag{1}$$

A facial biometric template for two aligned sets of landmarks is composed as a result of element-wise subtraction of their coordinates. However, the result values signify some

pixel distance between coordinates and thus depend on the image characteristics. To avoid that, we normalize the template elements to the constant perimeter $P_{sup}$. Due to the parameters of the employed active appearance model, the resultant descriptor includes 136 values.

Many other geometry-based descriptor implementations usually consider subsets on the landmarks or rely on hand-crafted features which are manually designed by selecting special relations within the active appearance model. Our implementation retains the geometric data in its entirety and gives a profound geometric description. However, the impact of discriminative power for different template elements is not equal and requires proper analysis or weighting which we achieve by learning methods.

To increase the template robustness against specific biometric distortion attacks (e.g., when the fake face image is warped to fit the geometry of the original image), texture features are also included in it. The texture descriptor in our method is based on the combined usage of HOG and LBP techniques. In order to define the image regions from which the features are extracted, we perform the segmentation of aligned input face image by ten characteristic semantic sections. The contours of the sections are depicted in Figure 2. The particular choice of the region's borders is based on intensive experiments of searching for a better selection. For each region, we extract eight HOG and eight LBP texture features with conventional computer vision tools. As a result of extracting sixteen features for each of ten regions, we get 160 texture components that are combined with geometric ones in a complete biometric template which includes $D\_size = 296$ features $\{d_i\}$.

### 3.2. PCA Template Compression

The designed template provides a holistic description of the face, which may include some redundancy. To eliminate it and make the system more compact, we employ compression techniques and evaluate the effect of template compression on the performance of validation. As an automatic approach for compressing the designed descriptor, we use the well-known principal components analysis (PCA) algorithm.

PCA is used for reducing the dimensionality of a template by projecting its elements onto a lower number of principal components, while at the same time maximizing the variance of the data.

### 3.3. Differential Template Verification

The process of document validation follows the differential scenario when the comparison is performed for two facial biometric templates extracted from this document. The first one $\{d\_test_i\}$ is extracted from the printed face photo, which is potentially counterfeited. Another template $\{d\_orig_i\}$, which is securely encoded with MRC, acts as a trusted reference. The match comparison decision signifies the genuineness of the tested document sample.

The superficial comparison indeed may be performed by applying the Euclidean distance metric (Equation (2)). This simple similarity score can be used to make the validation decision by comparison with the fixed threshold. However, different template elements can have different impact weight on the verification decision, which is not accounted for in this trivial linear estimation.

$$E = \sum_{i=1}^{D\_size} |(d\_test_i - d\_orig_i)| = \sum_{i=1}^{D\_size} |d\_sub_i| \tag{2}$$

Instead of tuning the similarity metric parameters manually, at this stage, we rely on the learning approaches. For such robust verification, we train a binary match/non-match classifier which is designed as a fully connected artificial neural network with a sigma activation function along with the network architecture.

As input, the classifier takes the absolute values of Its first layer receives the absolute result of element-wise subtraction of biometric templates $d\_sub_i$.

The final layer of the architecture includes a single node that outputs a scalar *S* in the range [0,1], where 0 corresponds to the ideal match decision. We train the classifier in the regression manner with the use of standard sequential back-propagation [26].

In order to normalize the values of network input and fit it better with the first layer activation function at the initialization stage, we introduce the coefficient *N* (see Equation (3)). In our experiments, the best optimization is achieved by setting $N = 0.015$ for geometrical template components and $N = 0.1$ and $N = 0.05$ for HOG and LBP texture components.

$$d\_inp_i = max(1, \frac{|d\_sub_i|}{N})\qquad(3)$$

## 4. MRC Application

The match-on-document scenario implies embedding the document sample with additional machine-readable data, which can be implemented in various ways. In our work, we follow the trend to employ computer vision tools to make the document readable with a digital camera. Such an approach may be conveniently implemented with a machine-readable code printed on the surface of the document (see Figure 3). For this work, we utilize the Graphic Code [27] that can be customized for security purposes.

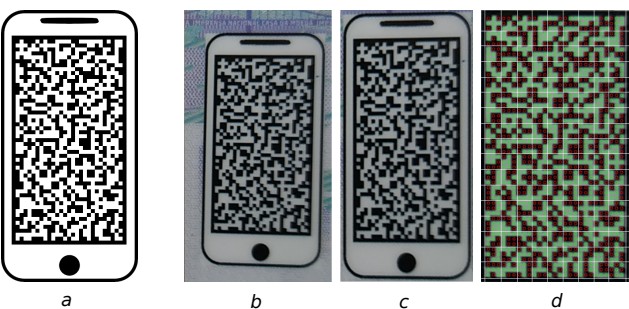

　　　　*a*　　　　　　　　*b*　　　　　　*c*　　　　　　*d*

**Figure 3.** Stages of MRC processing: (**a**) Generated MRC sample to be printed on the ID document; (**b**) detected MRC on the ID document printed sample; (**c**) frontalized MRC image; and (**d**) MRC message reconstruction [5].

We assume that the MRC sample for a particular biometric template is secured to store the trusted reference data for differential comparison, which is crucial for our approach. However, we refer to the detailed consideration of security issues to corresponding work [27]. We can summarize this discussion as follows.

During the process of creating the MRC sample, several layers of security and attack robustness can be introduced. These methods usually follow the symmetrical security approach, where various parameters of the system remain obscured and act as a key for both encryption and decryption. While the algorithm of assembling MRC remains open, the Graphic Code ensures the security performance by specifying its private internal parameters (e.g., unit cell characteristics and dictionary). To increase the security level, various cryptography methods over the data themselves may also be employed. For instance, to magnify the computation complexity of cryptanalysis, the message to be encoded may be encrypted to ciphered text. However, the usage of only these symmetrical methods leads to the overall risks, when compromising the application on a single portable device poses a threat to the full system.

This may be mitigated by adapting any asymmetric encryption approach. Indeed, to prove the originality of the printed MRC during the decoding process, it is required just to validate the document issuer's authority. To achieve that, the message with a biometric template can be protected with the digital signature which is a well-tested method for similar applications. This method requires the issuer to generate its private and public keys. The hash, which is extracted from the template data, is signed with a private key and added to the result message, which is encoded into the MRC. In that scenario, having the public key, the authority of the issuer of the document may be verified. The offline mode of

this verification can be maintained by uploading the public key of the issuer once during the initial installation.

### 4.1. Encoding

The Graphic Code allows an arbitrary choice of the outline image to ease the coordination with standards, which are usually applied to the appearance of the ID document. As an example in the scope of this work, we use one depicted in Figure 3a. The required alphabet defines the correspondence between $N$ (120 in this work) characters and various unit cells composed of $3 \times 3$ pixels. To encode the biometric template to the MRC, the result message is transferred into the alphabet space by quantization process. To compose the MRC instance, each character in the message is replaced by the corresponding pattern basing on the defined dictionary. The total amount of information $I$ that is carried by the encoded template results to $\sim$260 bytes, which is estimated by Equation (4).

$$I = K \cdot \log_2 N \tag{4}$$

Practical application may also require the encoding of some biographical data (ID card number and name) in addition to the biometric template to ease automatic document processing. To verify the correct decoding, a set of check digits is concatenated with the message. If any empty cells are left, they may be replaced with random non-dictionary unit cells.

### 4.2. Decoding

During the process of decoding, the captured image of the detected MRC is processed with conventional computer vision algorithms to achieve the properly aligned binary form suitable for decoding (Figure 3b,c). Next, the rectified image (Figure 3d) is aligned and examined to find unit cells defined in the dictionary for combining the result message.

The print/capture process introduces various distortions to the image of the MRC. Errors that occur due to various illumination conditions, reflection and MRC surface attrition may be detected with the use of check digits. We performed various tests with the various acquisition parameters and MRC deformations, to prove the overall robustness of the decoding algorithm. However, in practical applications, most of the inaccuracies can be compensated by processing the stream of frames captured by a digital camera.

## 5. Datasets

It is important to note that the deployment of a mobile face recognition system for document security purposes implies dealing with images that are captured by the portable digital camera from the physically printed documents.

As an example, some works directed on document scanning scenario (which is constrained with the absence of perspective transformations) introduce collected print-scan datasets to deal with that problem [28] or methods for generating such print/scan datasets automatically [29]. However, the last option usually can barely cover all aspects of illumination and acquisition distortions with various capture devices.

The document acquisition with a portable digital camera introduces even more variable noise to the captured images, which applies to the perspective inaccuracies and more severe lighting distortions.

Following the original purpose of protecting the ID document at the moment of its personalization, for all images from the chosen original dataset, we collected their print/capture digital copies trying to cover possible noise and distortion variations during the process. Acquired images were then automatically frontalized (see examples in Figures 4 and 5).

The differential verification with such an approach implies estimating the similarity between the original and the captured images in a single verification iteration. In order to retain such differential manner of the processing scenario, we labeled the sets of collected images similarly for original and captured ones.

A similar print/capture dataset (Print/Capture Aberdeen [30]) was obtained by Medvedev et al. [5]. We extend that approach to several other datasets that contain frontal face images complied with travel document standards [4] (Utrecht [30] and AR [31]), including various level of acquisition parameter variations. As a result, we obtain several print/capture datasets (https://github.com/visteam-isr-uc/trustfaces-template-verification (accessed on 30 June 2021)) which we call as follows :

- Print/Capture Aberdeen v2 (89 identities, 15 k captures);
- Print/Capture Utrecht (67 identities, 16 k captures); and
- Print/Capture AR (135 identities, 29 k captures).

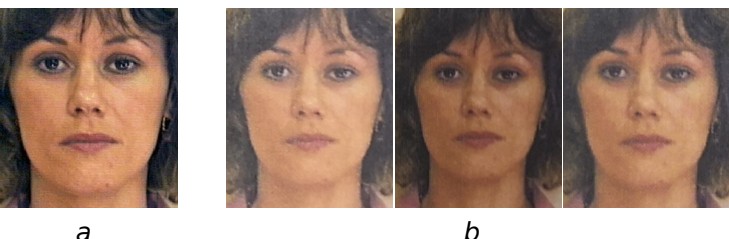

*a*       *b*

**Figure 4.** Example images from the Print-Capture Aberdeen dataset: (**a**) original digital image; and (**b**) captured photo image.

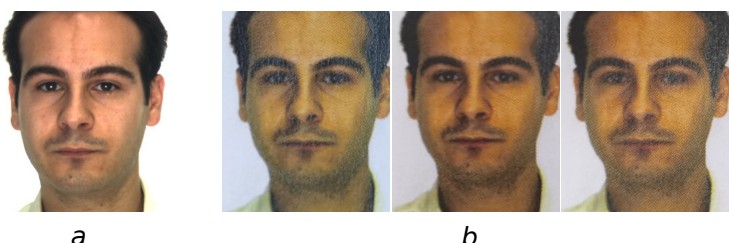

*a*       *b*

**Figure 5.** Example images from the Print-Capture AR dataset: (**a**) original digital image; and (**b**) captured photo image.

In the process of training on the templates extracted from such datasets, the network learns the proper weighting for the particular template components. At the same time, such an approach optimizes the further face verification process by learning existing irregularities related to printing, digital capture, misalignment and rectification process.

At the same time, practical deployment in the mobile application assumes handling the stream of frames from which only a few are selected for processing when the bad quality ones (over illuminated or occluded) may be skipped. Such practical details are usually important to be accounted in the early research stage. That is why one has to be careful choosing the strategy for selecting images to be included in the dataset, reducing ones that will likely be skipped during the deployment.

We achieve that by carefully designing a rectification process that is primarily directed on the eliminating perspective deformation of the document on the captured image.

### 5.1. Document Rectification

In a comparison with the automatic border control (ABC) scenario, the document validation with a mobile device deals with variations of the document alignment. That is why document rectification is usually a mandatory step in the validation pipeline. This operation allows obtaining frontalized and standardized images of the document for further processing and extracting data embedded in it.

This operation indeed may be performed in various ways. We assume that the ID document is flat and perform rectification by the trivial perspective transformation (Figure 6). The parameters of the transformation matrix are estimated by matching the detected features of the specific regions of the document [32]. Despite the fact that during

that process we require a specific document appearance, it is used only as a reference to achieve the main goal—frontalize the face images. This automatic process indeed also introduces additional warping noise to the result images.

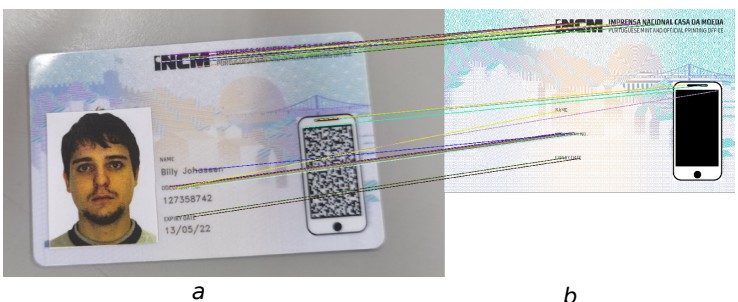

**Figure 6.** Rectification process with the featured detected by the document appearance: (**a**) captured document image; and (**b**) original card layout.

*5.2. Train and Test Protocols*

As stated above, the face recognition task in the target scenario is to verify the particular face image sample. That is why we define the training and testing protocols as follows. From the labeled sets of original and printed/captured images, we select pairs for extracting templates and computing their element-wise subtracts $\{d\_sub_i\}$. This set is also labeled in binary form, depending on their cross identity label. If images in the pair belong to a single identity, this pair is labeled as bona fide. Pairs with images from different identities are labeled as a biometric impostor attack.

In order to make the extracted data balanced in terms of the presence of match and non-match pairs, we significantly reduce the number of non-match ones to be included in the resulting protocol. This choice is randomized based on the overall dataset statistics. In our experiments, the result data were divided into train and test protocol parts with split proportions of 80% and 20%, respectively. The identities lists are disjoint in these two parts.

## 6. Results

In a recent work [5], we focused on the search for a better classifier architecture to optimize it in terms of the balance between efficiency and complexity. In this study, we mostly focused on estimating the impact of template compression on the overall system performance. We employed an ANN-based classifier with the following combination of hidden layer numbers: 300-400-200-100. The template was compressed by selecting a number of the first PCA components (compressed features) (Figures 7–10). To demonstrate the compression effect for each template part, we separately applied PCA to the geometric and texture sections. For geometric template, we first took 120 and 100 components. For texture template, we took 140 and 120 components (features).

With these settings, we performed intensive experiments and trained the set of classifiers for compressed collections of templates according to defined protocols. In each iteration, we defined the number of epochs equal to 20 and chose the best result at the end of the training.

As a metric for evaluating the performance, we estimated ROC (receiver operating characteristic) curves and computed their corresponding AUCs (area under curve) (see Figures 7 and 8 and Table 1).

We also estimated the performance with false acceptance rate and false rejection rate metrics (see Figures 9 and 10).

From the obtained results, one can see that the compression of the designed geometrical features does not significantly affect the performance. Indeed, this is related to the redundancy of defined descriptors as the locations of neighbor landmarks are highly correlated. At the same time, the drop in the performance with the compression of the texture features is more significant as they have much less redundancy. These features are extracted holistically from the semantic regions which do not intersect.

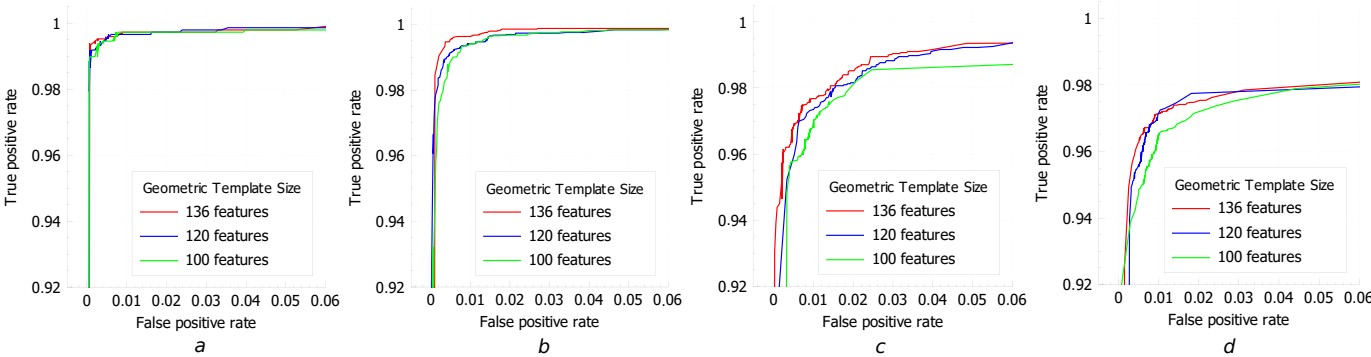

**Figure 7.** ROC curves of ANN classifier for various various compression of geometric features (PCA components): (**a**) Print/Capture Aberdeen; (**b**) Print/Capture Aberdeen v2; and (**c**) Print/Capture Utrecht; and (**d**) Print/Capture AR.

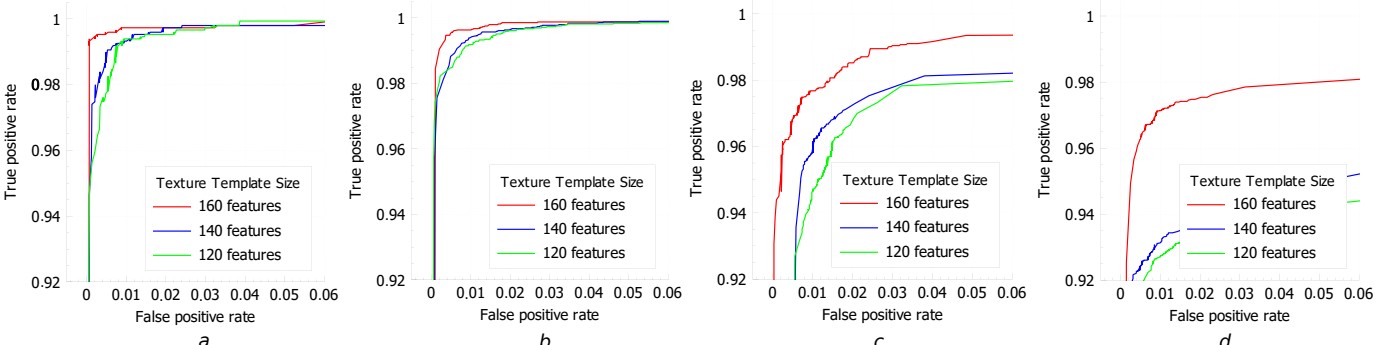

**Figure 8.** ROC curves of ANN classifier for various compression of texture features (PCA components): (**a**) Print/Capture Aberdeen; (**b**) Print/Capture Aberdeen v2; (**c**) Print/Capture Utrecht; and (**d**) Print/Capture AR.

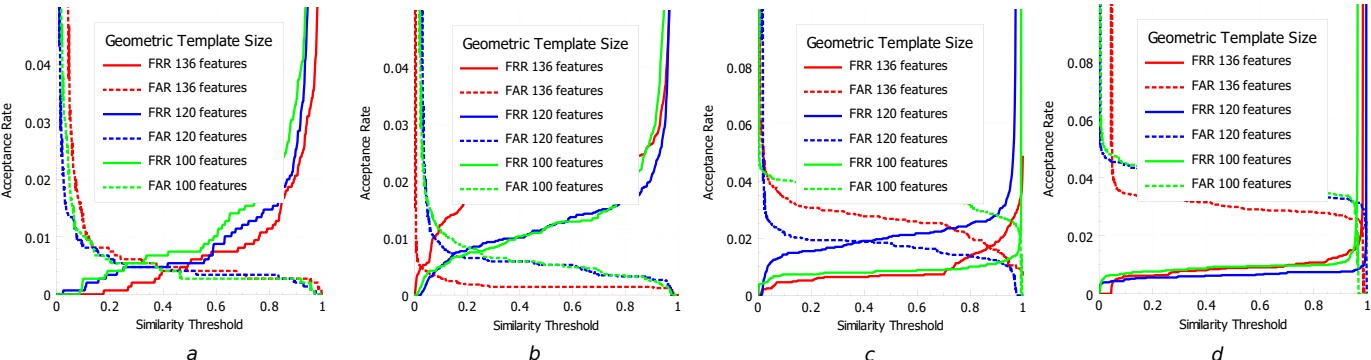

**Figure 9.** FAR/FRR of ANN classifier for various various compression of geometric features (PCA components): (**a**) Print/Capture Aberdeen; (**b**) Print/Capture Aberdeen v2; (**c**) Print/Capture Utrecht; and (**d**) Print/Capture AR.

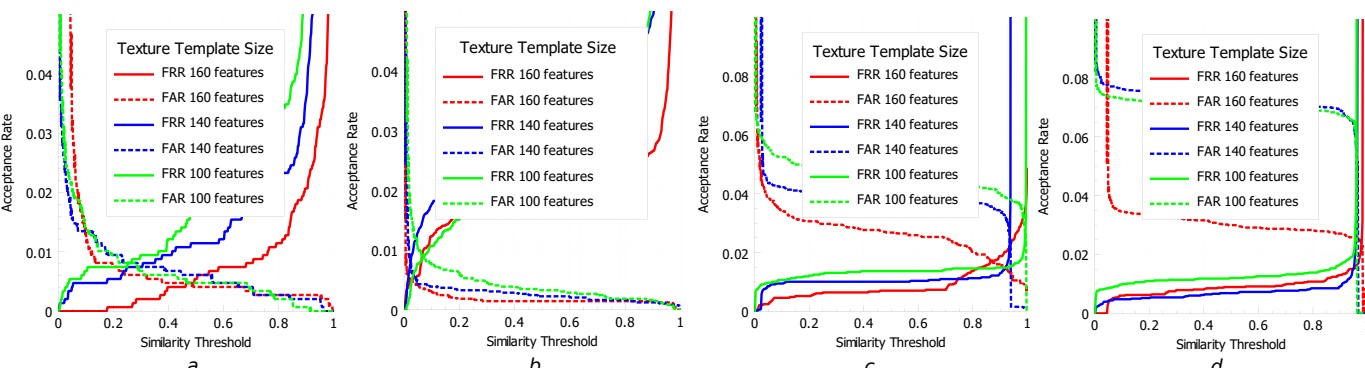

**Figure 10.** FAR/FRR of ANN classifier for various compression of texture features (PCA components): (**a**) Print/Capture Aberdeen; (**b**) Print/Capture Aberdeen v2; (**c**) Print/Capture Utrecht; and (**d**) Print/Capture AR.

**Table 1.** Performance characteristics (AUC/FNMR@FMR = 0.01/Equal Error Rate(EER)) of classifiers for various sizes of compressed templates and datasets.

| Template Size Geometric + Texture | Print/Capture Aberdeen | Print/Capture Aberdeen v2 | Print/Capture Utrecht | Print/Capture AR |
|---|---|---|---|---|
| 136 G + 160 T. | 0.999438/0.0027 0.0047 | 0.999179/0.0036 0.0048 | 0.99592/0.023 0.016 | 0.994624/0.029 0.023 |
| 120 G + 160 T. | 0.999465/0.0034 0.0047 | 0.999394/0.0059 0.0073 | 0.998035/0.027 0.018 | 0.992643/0.073 0.018 |
| 100 G + 160 T. | 0.999452/0.0027 0.0054 | 0.999277/0.0063 0.0077 | 0.995276/0.033 0.020 | 0.994577/0.078 0.024 |
| 136 G + 140 T. | 0.999262/0.0068 0.0074 | 0.998991/0.0061 0.0077 | 0.992228/0.044 0.024 | 0.987724/0.030 0.043 |
| 136 G + 120 T. | 0.999297/0.0075 0.0081 | 0.99912/0.0081 0.0087 | 0.991232/0.059 0.026 | 0.982557/0.038 0.047 |

The expected feature of the results is that the experiments on the different datasets demonstrate slightly different performance, which occurs due to the different level of illumination condition variations (exposure, relative position of light source and camera and applied shadows) during the process of their harvesting.

Another observation is that, depending on the template size, the parameters of the architecture (sizes of the hidden layer), which may be optimized to achieve the best accuracy, indeed depend on the template size (the size of the input layer). However, here we do not follow that suggestion, limiting ourselves to only estimating the compression effect with the fixed parameters of the setup.

## 7. Conclusions

This paper is devoted to the development of an efficient method for protecting ID and travel documents by augmenting them with a secured facial biometric template to be encoded in the machine-readable code. The approach is optimized for portable devices (e.g., smartphones) in terms of the CPU usage and solves the frontal face verification problem in the offline match-on-document scenario. Our demo application on an iPhone 7 is able to perform the complete card validation process (including detection, rectification, extracting templates and verification) in 0.2 s.

We introduce the improved facial biometric descriptor and estimate the effect of its compression on the performance of the system in various experiments.

As an additional contribution of this work, we introduce several print/capture datasets that may be useful for the research related to face recognition for mobile document security applications. They can be employed to analyze the robustness of face recognition algorithms to various distortions caused by the combined impact of printer and digital camera.

The overall results show the high performance of the developed method against biometric impostor attacks. At the same time, it may be customized with the use of biographical data or adapted for other biometric characteristics (such as fingerprints and iris). The method may be applied without sophisticated equipment, in a very cheap and convenient way. Our future work will be directed towards increasing the robustness of the developed facial template, more detailed analysis of the performance with a multi-fold approach and adapting deep learning techniques for the match-on-document scenario.

## 8. Patents

The results of our work on the project TrustFaces were published in the patent [33].

**Author Contributions:** Conceptualization and methodology, N.G. and L.C.; software, L.C.; investigation, data curation and writing—original draft preparation, I.M.; writing—review and editing, F.S.; and supervision, project administration and funding acquisition, N.G. All authors have read and agreed to the published version of the manuscript.

**Funding:** The authors would like to thank the Portuguese Mint and Official Printing Office (INCM) and the University of Coimbra for the support of the project TrustFaces.

**Institutional Review Board Statement:** Not applicable. The study involved the usage of the human face images, which were taken from the publicly available datasets.

**Informed Consent Statement:** Not applicable. The study involved the usage of the human face images, which were taken from the publicly available datasets.

**Data Availability Statement:** One of the results of presented work is a set of print/capture face datasets that are harvested with the use of publicly available images. (https://github.com/visteam-isr-uc/trustfaces-template-verification (accessed on 30 June 2021)).

**Conflicts of Interest:** The authors declare no conflict of interest. The funders had no role in the design of the study; in the collection, analyses, or interpretation of data; in the writing of the manuscript, or in the decision to publish the results.

## Abbreviations

The following abbreviations are used in this manuscript:

| | |
|---|---|
| MRC | Machine Readable Code |
| ANN | Artificial Neural Network |
| ABC | Automatic Border Control |
| ROC | Receiver Operating Characteristic |
| AUC | Area Under Curve |

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
