# Peer review of "Towards Facial Biometrics for ID Document Validation in Mobile Devices"

_applsci, doi:10.3390/app11136134_

Round 1

Reviewer 1 Report

The paper discusses automatic methods of compressing the developed face descriptors to make the biometric validation system more suitable for mobile applications.

Comments:

  1. The novelty of the paper is not clear. What do you mean by “consider methods”? Do you propose a new method or just apply the already known method and measure its performance? The novelty must be clearly stated in the Introduction section.
  2. The overview of related works is poorly structured and rather chaotic. Organize the analysis by the number of dimensions and by the totality of face visible: 2D face recognition vs. 3D face recognition, full face recognition vs. occluded/masked face recognition. Discuss the use of the parts of the face (such as ears) for facial biometrics. See, for example, “Towards Facial Recognition Problem in COVID-19 Pandemic” and “Secure ear biometrics using circular kernel principal component analysis, Chebyshev transform hashing and Bose–Chaudhuri–Hocquenghem error-correcting codes”. Also, you discussed machine learning methods, but did not discuss the deep learning methods, which are mainstream now.
  3. What facial landmark model did you use? How did you select facial landmarks as there are many suggested templates and alternatives in the literature? How does the set of landmarks you use compare with other sets of landmarks known from the literature, see for example, “Assessing Facial Symmetry and Attractiveness using Augmented Reality”.
  4. Explain the “combined usage of HOG and LBP techniques” in more detail. How from 68 landmarks you arrive at 160 texture components? Perhaps a figure explaining the structural composition of a feature vector would be helpful.
  5. How much principal component do you take from PCA?
  6. What is the size of the MRC matrix depicted in Figure 4? How many different codes it can hold?
  7. How do you select the sizes of neurons in hidden layers of your ANN? Perform and present the ablation study.
  8. The datasets you use are small (only 135 identities in the largest dataset). How is your approach scalable to thousands or millions of identities?
  9. Evaluate the performance of your biometric scheme in terms of False Accept Rate (FAR), False Reject Rate (FRR), and Equal Error Rate (EER).
  10. Evaluate the computational complexity of your method.
  11. How do your deal with the challenges of different facial expressions, pose variations, occlusion, aging, and facial hair? Discuss each aspect in detail.
  12. The conclusions should go beyond the summary of the work done. Support your claims with the main experimental results. How do your findings contribute to the research in the field of biometrics? Discuss future work.

Author Response

Dear Reviewer,

Thank you for your good suggestions and remarks. We have made several improvements to our work basing on your comments.

1. In our work we evolve the concept of the match-on document face verification, introduce several improvements to the recently developed template and present several print/capture datasets. Apply the PCA for compressing the biometric template. The corresponding comment was added to the introduction.

2. This work is focused on application of face recognition to document security with its specificity(frontal 2D faces without occlusion). We do not focus neither on considering 3D face recognition and its comparison with 2D face recognition, nor masked face recognition. 
We also do not focus on profound revision of CNN based methods since we develop our system for specific purpose and scenario (match-on-document) where computationally heavy deep learning  methods are overkill.

3. For detecting facial landmarks we used build-in opencv functionality which is a rather standard choice for detecting facial landmarks. However our method indeed can be expanded to any other of many available methods. Our particular choice is not the principal.

4. The features are extracted from the 10 characteristic semantic regions which are obtained as a result of segmentation with use of landmarks. Some clarification was added to the work. 

5. We do not specify exactly how many components we include in the final solution. Our goal was to demonstrate the effect of excluding components on Fig 9.   

6. The particular MRC implementation, which is given as an example, is inherited from our previous work [4] and consists of 9*17 unit cells. However this is not the only available choice and it is adjustable with Graphic code tools for particular needs. We added a clarification regarding the size of the template to be encoded which takes ~260 bytes.

7. Indeed that study was made in our initial work [4].

8. We didn't address this question due to the limited choice of available ID document compliant face datasets. However this is a good idea for a study in future works.

9.The results were additionally represented  with FNMR@FMR=0.01 metric.

10.  Deployment of full approach including the card and face detection, rectification, extracting templates (from face image and decoded MRC) was running with 4-5 frames per second in an application on Iphone 7.

11. We stated that our method is not intended to cover these aspects of face recognition. Our goal is to protect a particular instance of the ID document that contains a single. That means that the new document issued - the new template is generated which is supposed to belong to this particular document. With this approach covering these aspects of face recognition is unnecessary.

12. In this work we adapt the face recognition to the concept  of match on document verification, introduce several print/capture datasets which may be useful for the research related to the similar systems for mobile devices. Clarification regarding the future work is added.

Reviewer 2 Report

The work is well structured and easy to follow. There are a lot of minor language issues that should be resolved by a thorough proofread, e.g., on page 1:
l. 14 "is" => "are"
l. 23 "implies," => "implies"
l. 29 "what" => "which"
l. 31 "numbers" => "a number of"
...

The authors should provide a more detailed comparison to their prior work in [4] to justify general overlap and to detail the significance of this follow-up work. Also, please cite the original source of repurposed figures where appropriate.

It seems the Aberdeen dataset is not referenced.  

It is not entirely clear how the print/capture generation from the source through the manipulation process via a trained model operates - will the model be publicly available? How many slightly different variations are there for each individual image? Are the different image versions close to one another? In that case this could give a false sense for the later results through very minor variations, but at a very large count.

Were multiple train-test splits (folds) performed? 

What are the differences between Aberdeen and Aberdeenv2?

Is there a reason not to combine all face images as might be needed in a real-world scenario?

In general, what is an example size of data that needs to be stored in the code on front of the card?

Author Response

Dear Reviewer,

Thank you for your good suggestions and remarks. We have made several improvements to our work basing on your comments.

- The language issues were revised and corrected.

- The better explanation of the differences with the previous work is included in the introduction. The figures were properly cited.

- The proper reference to the Aberdeen dataset was added.

- The goal of collecting the datasets was indeed to cover variations mostly related with exposure of the document in terms of relative position of lighting source and the camera which is crucial if the real deployment.  At the moment we don't intend to publish the model but some appropriate code may be published.

- Multiple train-test folds were not performed however this is a good idea for the follow up work.

- Aberdeen and  Aberdeen_v2 are similar however they are different in the size of collected images. Also the Aberdeen was collected with only Iphone6  when the  Aberdeenv2 was collected with Iphone6 + Huawei P20. Summing up the Aberdeen_v2 includes more acquisition variances which is seen on the performance results.

- Indeed joining the datasets will make sense in the development of a system for deployment. However we wanted to estimate and demonstrate the difference between the datasets. Training on the complete dataset is a good idea for future work. 

- The complete size of the uncompressed template consists of 296 symbols. With the alphabet size 120 the total amount of information results in ~260 bytes. We didn't specify many of the parameters since we follow their definition from our previous work [4] and here we tried to focus on different aspects. Some clarifications are added to the corresponding  sections.

Round 2

Reviewer 1 Report

My comments were addressed in part only. Therefore, a new round of revisions is required.

Comments:

  1. Explicitly state the novelty and contribution of this paper with respect to previous works, especially with respect to the previous work of the authors (ref. [4]). If this is an extended paper from the conference, it must be clearly stated. The inclusion of texture components in the biometric facial templates is not new and has been done many times before. The article uses very old methods such as HOG and LBP. The state-of-the-art methods for face-based document identification use deep learning methods now.
  2. It is still not clear to me how many principal components the authors take from PCA. These details should be given to allow replicability of the proposed method and its details.
  3. Add a figure (workflow) or an algorithm explaining how a feature vector is composed.
  4. Evaluate the performance of your biometric scheme in terms of False Accept Rate (FAR), False Reject Rate (FRR), and Equal Error Rate (EER).
  5. Figure 1 is taken from ref. [4] (Fig. 1). Remove or replace to avoid copyright violation.
  6. Figure 3 is taken from ref. [4] (Fig. 3). Remove or replace to avoid copyright violation.
  7. Language should be improved. There are many spelling errors and typos such as “Pefromance”.

Author Response

Dear Reviewer,

Thank you for your good suggestions and remarks. We have made several improvements to our work basing on your comments.

  1. We believe that the novelty is clearly stated in the introduction. The particular paragraph is highlighted.  We added the comment that this paper is the extended paper from the conference. Indeed the texture description methods were used for the face recognition before, however in our work we revised and combined these techniques and optimized its usage to mobile devices. We motivate the avoidance of deep learning methods at this stage of our work by their computational complexity in applications to portable devices, however we plan to study this tool in future studies.
  2. The number of  PCA components were named “features of the compressed template” which probably caused misunderstanding. We investigated the effect of template compression by excluding a number of last PCA components (compressed features). We investigated compression of  geometrical features and texture features separately. Several clarification comments were added.  
  3. The required figure is added instead of 2 and 3.
  4. We expected that the FNMR@FMR metric, which is more standart, could cover the lack of the paper.  Following the reviewer’s comment we have also added the required  FAR/FRR metric to the work.
  5. Figure 1 is replaced.
  6. Figure 2 and 3 are replaced with the new figure according to the comment.
  7. Additional revision of the text was performed. Errors and typos were corrected.

Reviewer 2 Report

The authors addressed some of the main concerns. This reviewer still contemplates if a multi-fold approach would yield more realistic and detailed insights into the performance of the algorithm through averages and confidence intervals that could be obtained when not arbitrarily relying on a single train/test split for the evaluation. I would recommend to at least note this shortcoming in the narrative and conclusion as future work.

Author Response

Dear Reviewer,

Thank you for your good suggestions regarding the multi-fold approach. We followed your recommendation and added this comment to the conclusion.

Round 3

Reviewer 1 Report

My previous comments were not fully addressed. Therefore, another round of revisions is required.

  • I can not find the statement on the novelty of your paper. You claim that “revise the implementation of the facial biometric template and modify it by including texture components”. Can you enlist what revisions were done? The inclusion of texture into the facial biometric templates is not new and has been done many times before.
  • Line 190: “The computational complexity requirements of deep learning approaches are still high” – you have used the high computational complexity of existing solutions as a motivation for your research. However, you did not present the evaluation of the computation complexity of your approach.
  • Line 210: the authors failed to mention CloudWalk and several other industrial solutions.
  • I still can not find the number of PCA components you used. Clearly state in the text the number of PCA components used and the amount of variability (%) they captured.
  • Line 259: “the result descriptor includes 136 values”. In Figures 7-9, you have used a various number of features: 136, 120, 100, 160, 140, 120. What is the difference between these feature sets? What is their composition? Present a table comparing and explaining the components of various feature sets used in your research.
  • Section 5.1: the rectification process requires that you have the document template for validation. Is it feasible in real-world conditions (such as border control at airports) to have document templates from every world entity issuing identification documents?
  • Table 1: the achieved results are well-below industry results obtained on much larger datasets, see https://pages.nist.gov/frvt/html/frvt1N.html . The authors failed to discuss it.
  • Table 1: add False Positive Identification Rate (FPIR).
  • Line 464: “acquisition parameter variations”. Explain in more detail what are the acquisition parameters and how they vary during the acquisition process.
  • Line 473: “approach is optimized for portable devices” – I did not find the description in the paper of how it is optimized for smartphones. What is optimized? CPU usage? Memory use? Power consumption? Detailed explanations should be added.
  • Line 477: “extensive experiments.” – the experiments were not extensive as their results fit in a single table.

Author Response

Thank you for your good suggestions and remarks. We have made several improvements to our work basing on your comments.

1.We believe that the novelty is clearly stated in the introduction. The main revision we do is the development and including texture descriptor. Also there are several minor ones which are related to practical implementation issues and does not contribute significantly to the discussion. To satisfy the comment we removed the word “revise”.The particular implementation of the texture descriptor, that we use, is novel and is based on our experiments in the search of a better implementation. 

2. Computational compactness of our method is proven with successful deployment of the method on mobile devices. For instance it was running with 4-5 frames per second in an application on Iphone 7. We added the clarification comment in conclusion. 

3. From the big list of various face recognition solutions we selected ones that better suit the match-on-document scenario, which we follow in our approach. We consider listing all available face recognition industry solutions inappropriate for this work. 

4. The clarification of the number of PCA components (compressed features) is added to the Sec. 6, I paragraph. The understanding of the difference between these feature sets and their composition is straightforwardly followed from the PCA concept - as a compressed template we take the number of first PCA components (120, 100 geometric components and 140, 120 texture ones ).  We also believe that this simple and clear issue does not require the representation in the form of a table.

5. This comment is addressed in 4. 

6. Indeed the rectification we use requires a partially known appearance. It will be really challenging to apply it for “every world entity issuing identification documents”. However in practice it may be enough for particular environments (like within the company/institution). Indeed this process was described mainly to demonstrate how the datasets were collected, since it will be conceptually similar in other match-on-document applications for smartphone devices.  

7. Our compact mobile approach does not intend to compete with the methods developed for desktop applications with rather light computational restrictions. Also the comment refers to the 1-N authentication page, which is inappropriate for comparison with the verification scenario considered in the paper. 

8. The considered approach follows “match-on-document” verification which is a particular case of 1-1 verification. We do not consider applications to 1-N scenario. That is why considering FPIR metric which corresponds to 1-N authentication are out of the scope of the approach. 

9.We better clarified the parameters according to the comment.

10. We added clarification in regards to this comment.

11. Indeed the number or experiments that were performed with different architectures and compression levels is several times bigger that represented in the paper. We represented the results in a compact form. However, to satisfy the comment we removed the word “extensive”.